# Genome-Wide Computational Prediction and Analysis of Noncoding RNAs in *Oleidesulfovibrio alaskensis* G20

**DOI:** 10.3390/microorganisms12050960

**Published:** 2024-05-10

**Authors:** Ram Nageena Singh, Rajesh K. Sani

**Affiliations:** 1Department of Chemical and Biological Engineering, South Dakota Mines, Rapid City, SD 57701, USA; ram.singh@sdsmt.edu; 22-Dimensional Materials for Biofilm Engineering, Science and Technology, South Dakota Mines, Rapid City, SD 57701, USA; 3Data Driven Material Discovery Center for Bioengineering Innovation, South Dakota Mines, Rapid City, SD 57701, USA

**Keywords:** biofilm formation, noncoding RNAs, *Oleidesulfovibrio alaskensis* G20, sRNAs, sulfate-reducing bacteria

## Abstract

Noncoding RNAs (ncRNAs) play key roles in the regulation of important pathways, including cellular growth, stress management, signaling, and biofilm formation. Sulfate-reducing bacteria (SRB) contribute to huge economic losses causing microbial-induced corrosion through biofilms on metal surfaces. To effectively combat the challenges posed by SRB, it is essential to understand their molecular mechanisms of biofilm formation. This study aimed to identify ncRNAs in the genome of a model SRB, *Oleidesulfovibrio alaskensis* G20 (OA G20). Three in silico approaches revealed genome-wide distribution of 37 ncRNAs excluding tRNAs in the OA G20. These ncRNAs belonged to 18 different Rfam families. This study identified riboswitches, sRNAs, RNP, and SRP. The analysis revealed that these ncRNAs could play key roles in the regulation of several pathways of biosynthesis and transport involved in biofilm formation by OA G20. Three sRNAs, *Pseudomonas* P10, Hammerhead type II, and sX4, which were found in OA G20, are rare and their roles have not been determined in SRB. These results suggest that applying various computational methods could enrich the results and lead to the discovery of additional novel ncRNAs, which could lead to understanding the “rules of life of OA G20” during biofilm formation.

## 1. Introduction

Noncoding RNAs (ncRNAs) are untranslated short transcripts that play essential roles in many cellular processes. A subgroup of ncRNAs is called small noncoding regulatory RNAs (sRNAs). sRNAs are short (50–500 nt) and can be mainly two types: (i) trans-sRNAs and (ii) cis-sRNAs. Trans-sRNAs are synthesized from intergenic regions (IGRs) of the genome and cis-sRNAs are encoded by the antisense strand (Figure 1). Trans-sRNAs regulate the target-gene expression, interacting with the ribosome-binding region, while cis-sRNAs can pair with target mRNAs to repress or modulate target-gene expression. A variety of sRNA-based regulatory (RNA–RNA interaction) mechanisms have been identified for controlling the expression of metabolic pathways [1], stress responses [2], and pathogenesis [3]. ncRNAs play significant roles in post-transcriptional regulatory strategies [4,5]. ncRNAs not only regulate regulatory pathways e.g., quorum-sensing circuits and signaling molecules [6] and reversible and irreversible genetic events (e.g., methylation of the genome) [7], but their stochastic variations (variation in abundance of ncRNAs) during transcription and translation [8] result in a change in protein levels.

ncRNAs such as riboswitches, T boxes, and sRNAs are the RNA molecules that play an important role in regulation of gene expression. sRNAs were first reported in *Escherichia coli* and recognized as key transcriptional regulators [9], due to their swift response to modulate mechanisms in the bacteria [10]. Studies show that ncRNAs regulate biofilm formation in bacteria such as *P. aeruginosa* [11], *B. cepacia* complex [11], *E. coli* [12] and *S. enterica* Typhi [13]. In *E. coli*, researchers have reported that sRNAs such as the multicellular adhesive (McaS) sRNA bind to mRNAs and modulate synthesis of curli and flagella, which leads to downregulation and upregulation of these cell surface structures, respectively. McaS also regulates the synthesis of the exopolysaccharide β-1,6 N-acetyl-D-glucosamine (PGA) by binding the global RNA-binding protein CsrA. The inactivation of McaS RNA leads to compromised CsrA binding, PGA regulation, and biofilm formation in *E. coli.* Studies have shown that Hfq-binding sRNAs play key roles in the regulation of biofilm formation-related processes such as the expression of flagella, curli fibers, and colanic acid (part of the exopolysaccharides), cellulose, and PGA in *E. coli*. sRNAs modulate gene regulation of *flhDC* mRNA responsible for flagellar synthesis, repressed by sRNAs (ArcZ, OmrA, OmrB, and OxyS) and activated by McaS [14,15].

In enteric bacteria, RydC reduces the expression of the *csgD*, a key factor for adhesion and biofilm formation. For example, in *E. coli* and *Salmonella enterica*, expression of RydC inversely affects biofilm formation [12]. The expression of sRNA SrbA was found to be 45-fold greater in biofilm than in the planktonic exponential phase in *P. aeruginosa* [16]. Orell et al. reported 29 ncRNAs expressed in biofilms of the thermophilic archaeon *Sulfolobus acidocaldarius* [17]. tRNAs and rRNAs are also ncRNAs that interact with mRNAs in the synthesis of proteins.

Biofilms are complex ecological niches formed by microorganisms on various surfaces using self-synthesized exopolymeric substances (exopolysaccharides, proteins, lipids, nucleic acids, etc.). The biofilm matrix mainly consists of polysaccharide intercellular adhesin, extracellular and cell surface-associated proteins, and extracellular DNA (eDNA). Thus, biofilm formation by microorganisms is a versatile and adaptive feature and its phenotypic variations could be result of genome regulation [18]. A recent study confirms that sRNAs are also involved in iron regulation in biofilms of *Aggregatibacter actinomycetemcomitans* [19]. In another study, a regulatory sRNA (PrrF1/F2) was shown to be involved in eDNA release and extracellular matrix formation [20] in *P. aeruginosa*.

Sulfate-reducing bacteria (SRB) gain energy through dissimilatory sulfate reduction [21,22]. SRB use sulfate as a terminal electron acceptor, reducing it to hydrogen sulfide, and thrive in various natural habitats (e.g., freshwater sediments and salty marshes), deep subsurface sites (oil wells and hydrothermal vents), and in an industrial setting. SRB can induce fouling, corrosion, and pipeline clogging by forming biofilms. SRB leads to enormous financial losses (>4 billion USD) in the USA by producing microbially induced corrosion (MIC) [21]. Biofilms are complex multilayer structures of bacterial cells where the initial layers in contact with the metal perpetuate anoxic ambience, promoting growth of SRB [23]. The mechanisms of biofilm formation by SRB involve several pathways governed by many genes that are regulated and differently expressed. The SRB biofilms are composed of extracellular proteins with a minimal exopolysaccharide content [24,25] called extrapolymeric substances (EPS). It has been reported that SRB biofilms vary in expression of transcripts and proteins related to categories such as carbon and energy metabolism, amino acid metabolism, stress response, proteases, and ribosomal proteins [25,26]. Biofilm matrix formation is controlled and regulated by the quorum-sensing (QS) system [27], a cell–cell communication system. Autoinducers initiate reaction to produce more biofilm exceeding the cell density to a threshold by modulating expression of genes [28]. Simultaneously, the dynamic niche of biofilm involves stress response regulators to combat stress conditions and stimulate changes in gene expression that help SRB to survive [25,29]. Considering the complex organizational structure and functional mechanisms of biofilms, it is required to understand the “rules of life” of SRB to avoid biofilm formation. The mechanism of biofilm formation comprises four stages; (1) initial attachment of cells, (2) proliferation, (3) maturation, and (4) dispersal (Figure 2). These stages involve expression of various genes and their regulation. Though many studies are available on gene and protein expression during biofilm formation, the regulome (ncRNAs) of SRB biofilms is yet to be explored. A knowledge gap exists regarding the regulome of OA G20 (as shown in Figure 2 with a question mark). This study aimed to fill this knowledge gap by predicting and identifying the ncRNAs transcribed from the OA G20 genome and their association with biofilm formation. Since this was the first attempt to predict and identify ncRNAs in the genome of OA G20, we decided to report all the ncRNAs identified. Mainly, riboswitches such as the TPP riboswitch, FMN riboswitch, c-di-GMP riboswitch, and SAM riboswitches are directly linked to the sulfur metabolism and mechanisms of biofilm traits.

## 2. Materials and Methods

### 2.1. Prediction and Identification of the ncRNAs

To achieve the aim of prediction and identification of ncRNAs from the genome of OA G20, three different approaches were adapted to predict and identify ncRNAs from the genome of OA G20. In the first approach, a FASTA (.fa) file of the genome sequence (NC_007519.1/CP000112.1) of *Oleidesulfovibrio alaskensis* G20 was downloaded from NCBI. The whole-genome sequence was split in multiple FASTA files of 7 kb using faSplit v377 (https://github.com/gpertea/gsrc/blob/master/scripts/fasplit (14 December 2023)). These multiple FASTA files were used to predict and identify ncRNAs via batch search in the Rfam server [30]. The Rfam search applies different computational methods (e.g., cmsearch, nhmmer) and uses a database (RNACentral) to search and identify genome-wide distribution of ncRNAs in the genome of OA G20. Rfam and RNACentral [31] are RNA family databases for ncRNAs.

In the second approach, the genome of OA G20 (NC_007519.1/CP000112.1) was selected from the Rfam server, and predicted ncRNAs were selected for further analysis. In the third approach, Proksee genome annotation and a visualization server were used to predict and identify ncRNAs and their distribution in the OA G20 genome. The genome sequence (NC_007519.1/CP000112.1) from the NCBI genome server was uploaded for analysis. Proksee utilizes multiple tools for the annotation of the genome, including Prokka, and ncRNAs were predicted by cmsearch [32]. The software and tools used for analysis are given in Table 1.

### 2.2. Secondary Structures of ncRNAs

The secondary structures of the predicted ncRNAs were visualized using the visualization server R2DT [33]. The Rfam database contains RNA-sequence families of structural RNAs, including ncRNA genes and cis-regulatory elements. Each RNA family was searched and represented by multiple sequence alignment using a covariance model (CM). The CM includes an algorithm where templates are used as a reference structure prediction. Insertions, deletions, and repositioning of nucleotides was based on structural context by Traveler software v3.0.0 [34]. Infernal v1.1 [35] predicts ncRNAs using a profile hidden Markov model (HMM) scheme [36]. R2DT utilizes approximately 4000 templates for RNA secondary structures. tRNA nucleotide numbering was done using Sprinzl scheme [37]. R2DT employs multiple tools such as the Comparative RNA website (for small rRNA subunit and 5S rRNA templates) [38], Ribovision (for large rRNA subunit; LSU) [39], GtRNAdb (for isotype-specific tRNA templates) [40], RNAse P database (RNase P templates) [41] and Rfam. Prediction algorithms seek similarity in sequence and possible secondary structure features. Different templates from different sources were used for secondary structure prediction, such as for 5S_rRNA (*Empedobacter brevis* template from Comparative RNA website; CRW), c-di-GMP using c-di-GMP-I-GGC, LSU_rRNA by Rfam using BS_LSU_3D (RiboVision), SSU_rRNA using template EC_SSU_3D (RiboVision). Templates for secondary structures for 6S RNA, cobalamin, *Pseudomonas* P10, tmRNA, DDE18215 (hammerhead_II template), and sX4 sRNA were used from Rfam. For rnpB sRNA secondary structures, the template was RNAseP_b_N. Meningitidis-Z2491_JB and bacterial-Sec (B_Ser) were used as template tRNA_sec from GtRNAdb.

## 3. Results

OA G20 is a well-studied model SRB for MIC due to biofilm formation [21,42,43]. The OA G20 genome has 3257 CDSs and 66 tRNA genes (Table 2).

Three different approaches were used to predict and identify the genome-wide distribution of ncRNAs in OA G20 [44]. The first approach resulted in 34 ncRNAs, which were classified into 5 classes (tRNAs, rRNAs_bacteria, rRNAs_archaeal, rRNAs_eukaryotes, and riboswitch). These 34 ncRNAs (Appendix A) were further categorized (Table 3) into 14 family types: (i) tRNAs, (ii) tRNA-Sec, (iii) thiamine pyrophosphate riboswitch (THI element), (iv) S-adenosylmethionine (SAM) riboswitch (S-box leader), (v) glycine riboswitch, (vi) cobalamin riboswitch, (vii) 5S_rRNA, (viii) SSU_rRNA_bacteria, (ix) LSU_rRNA_bacteria, (x) SSU_rRNA_Archaea, (xi) LSU_rRNA_Archaea, (xii) SSU_rRNA_Eukarya (xiii) LSU_rRNA_Eukarya, and (xiv) SSU_rRNA_microsporidia. In the case of riboswitches, our analysis identified five riboswitches: two copies of the thiamine pyrophosphate (TPP) riboswitch (THI element) and one copy each of the SAM riboswitch (S-box leader), glycine riboswitch, and cobalamin riboswitch. Eight types of rRNA were identified: three from bacteria (5S_rRNA, SSU_rRNA and LSU_rRNA), two from Archaea (SSU_rRNA and LSU_rRNA), two from Eukarya (SSU_rRNA and LSU_rRNA) and one SSU_rRNA from microsporidia. The similarity of predicted rRNAs with eukaryotes and archaea was surprising, as OA G20 is a bacterium. A total of 21 tRNAs were predicted and identified including a single gene for tRNA-sec. The genome annotation showed 66 tRNA genes in the OA G20 genome (Table 2). tRNA-Sec (selenocysteine transfer RNA) is unique and plays a key role in the biosynthesis of L-selenocycteine (Sec or U). The identified tRNA-Sec of OA G20 has conserved and variable regions compared to other SRB, such as *Desulfolutivibrio* 92.55%, *Desulfovibrio sulfodismutans* 90.43%, and *Desulfovibrio fairfieldensis* 71.53%.

Identification of ncRNAs in OA G20 applying the second approach resulted in 93 sequences (ncRNAs). These genes were also classified (Appendix A) into 16 family types: (i) tRNA, (ii) 5S_rRNA, (iii) SAM_riboswitch, (iv) cobalamin_riboswitch, (v) FMN_riboswitch (FMN element), (vi) TPP_riboswitch (THI element), (vii) glycine_riboswitch, (viii) cyclic di-GMP-I_riboswitch (c-di-GMP_riboswitch), (ix) bacterial small signal recognition particle RNA (bacterial small SRP_RNA), (x) 6S/SsrS_RNA, (xi) bacterial large subunit ribosomal RNA (LSU_rRNA_bacteria), (xii) bacterial small subunit ribosomal RNA (SSU_rRNA_bacteria), (xiii) STnc490_Hfq binding RNA, (xiv) Pseudomonas sRNA P10, (xv) sX4, and (xvi) tRNA. Analysis revealed four copies each of 5S_rRNAs, SSU_rRNA_bacteria and LSU_rRNA_bacteria. Of eight identified riboswitches, TPP riboswitch had two copies (Appendix A), cobalamin riboswitch two copies (Appendix A), and there was one copy each of glycine riboswitch, FMN riboswitch, SAM riboswitch, and c-di-GMP-I riboswitch (Appendix A). In the second approach, we identified unique ncRNAs that were not predicted in the first approach, such as *Pseudomonas* P10 (two copies), STnc490, 6S, and bacterial SRP_RNA (Appendix A). The analysis identified 67 tRNAs. Further in-depth analysis of the tRNA sequences identified another class of ncRNA: sX4 (RF02223), a proteobacterial sRNA. Seven sequences (Appendix A and Appendix A) were identified for sX4 (OAg20_sX4a, OAg20_sX4b, OAg20_sX4c, OAg20_sX4d, OAg20_sX4e, OAg20_sX4f, and OAg20_sX4g). Therefore, in the second approach, we identified 16 families of ncRNAs in OA G20 (compared to 14 families in the first approach). This approach resulted in the identification of a greater number of ncRNAs, such as cobalamin riboswitch (second copy) FMN riboswitch, c-di-GMP-I riboswitch, P10, STnc490, sX4, 6S, and bacterial small SRPs, which were not identified via the earlier method. This method also precisely predicted and identified rRNAs for bacteria and did not show mispredictions as in approach 1.

Considering the improvement in the identification of new ncRNAs in the second approach, a third approach was adopted. A total of 92 ncRNAs (including 66 tRNAs) were predicted and identified (Figure 3) using Proksee, a genome annotation and visualization server (https://proksee.ca; (14 December 2023). Results of predicted ncRNAs are given in Appendix A and Appendix A. The identified ncRNAs were classified (Appendix A) into 16 families. Most of the identified ncRNAs were the same as identified in approach 2, except for three new predicted ncRNAs (Table 4): rnpB (bacterial RNase P class A), hammerhead ribozyme (type II), and (iii) tmRNA (Appendix A). The copies of rRNAs and riboswitches were the same as those identified in the second approach.

There were 66 tRNAs identified, including one tRNA-Sec. Another ncRNA identified via this approach was tmRNA. This approach identified ncRNAs that were not found in previous approaches, such as bacterial RNase P, hammerhead type II, and tmRNA (Table 4). Approach 3 identified an equal number of ncRNA families (16) as approach 2. Approach 1 identified 14 ncRNA families, but it falsely assigned similarity to eukaryotic and archaeal rRNAs (five families); therefore, it identified only nine ncRNA families that belong to bacteria. A total of 23 ncRNAs were common from approaches 2 and 3. Approach 2 predicted eleven ncRNAs that were not predicted in approach 3, and there were three ncRNAs unique to approach 3. In total, over all three approaches, a total of 37 ncRNAs were identified (excluding tRNAs) (Table 4). These 37 ncRNAs were distributed in 18 Rfam families.

### Secondary Structures of ncRNAs of OA G20

The secondary structures of ncRNAs were predicted using reference templates. The results showed that nucleotides were folded and classified into four categories: (i) same as template, (ii) modified compared to template, (iii) inserted nucleotides, and (iv) repositioned compared to the template. These secondary structures have shapes due to sequence folding and complementary hydrogen bonds and hairpin loops. The hairpin structures are called “stems”, and bulbs at the end of stems are called “loops”. The sequence strings or bulbs joining two or more stems are known as “joints”. Secondary structures have primary and secondary (hydrogen) bonds between nucleotides, which give shape and stability to the structures.

Riboswitches are important part of ncRNAs identified in the current study. The two identified TPP riboswitch sequences have variation in secondary structure in terms of all four categories mentioned. OA G20 TPP_riboswitch structures, OAg20_*TPP*a and OAg20_*TPP*b both have 27 secondary bonds. In terms of the nucleotide arrangement structure, OAg20_*TPP*a has 27 nucleotides and OAg20_*TPP*b has 29 nucleotides arranged as template (conserved sequences). OAg20_*TPP*a has seven and OAg20_*TPP*b five repositioned nucleotides. The structure OAg20_*TPP*a has six and OAg20_*TPP*b seven inserted nucleotides. There were other variations in nucleotide compositions of the loops, stems, and 3′-end and 5′-end, but these were conserved at important sites (Appendix A). OA G20 genome has two cobalamin riboswitches, represented as OAg20_cola and OAg20_colb. The structure OAg20_cola has 43 and OAg20_colb 39 secondary bonds and 42 and 46 conserved nucleotides, respectively. The structure OAg20_cola has seven nucleotides and four nucleotides and OAg20_colb has three nucleotides and five nucleotides inserted and repositioned. Both structures have variations in joint loop J3–4–5–6. Another distinct change was observed in loop 4, where structure OAg20_cola has eleven nucleotides, with four inserted, and OAg20_colb has only eight nucleotides. Loop 4 was observed in opposite directions in both structures. The secondary structure analysis of the glycine riboswitch showed that the structure has 31 nucleotides conserved and arranged as template and 22 secondary bonds. The structure has 22 inserted nucleotides in an L3 loop. The FMN riboswitch regulates mRNAs that encode for flavin mononucleotide biosynthesis and transport proteins. OA G20 has one sequence/gene for the FMN riboswitch (OAg20_fmn). The OA G20 FMN riboswitch’s secondary structure has six stems (P1–P2–P3–P4–P5–P6) and five loops (L2–L3–L4–L5–L6). The secondary structure has 28 secondary bonds. The structure has 28 inserted nucleotides in L3, 4 nucleotides in L4, and 1 nucleotide in L6. There were two repositioned nucleotides: one (U) in L3 and one (U) in L6. The OA G20 genome has one gene for the SAM-I riboswitch. The secondary structure of the OA G20 SAM-I riboswitch consists of five stems (P1–P2–P3–P4–P5) and three loops (L3–L4–L5). It has 31 secondary bonds and 41 conserved nucleotides. Two loops with inserted nucleotides L3 (U) and L5 (G and U) were also observed. There were three nucleotides (two (G, G) in L3 and one (A)) repositioned in P4. The OA G20 genome contains a single-copy gene for the c-di-GMP-I riboswitch. Secondary structure analysis revealed three stems (P1–P2–P3) and one loop—L3. The structure has 18 nucleotides as conserved and 18 secondary bonds and one repositioned nucleotide (G).

The OA G20 genome has another SRP RNA ffs. The secondary structure of ffs has 14 conserved nucleotides and 33 secondary bonds. The structure has one inserted (G) nucleotide. The 6S RNA is a small prokaryotic ncRNA that has a single copy in OA G20. Its typical secondary structure includes a big central “loop” edged by elongated double-helical arms. The OA G20 6S RNA secondary structure revealed that it has two large middle loops. Loop 1 is between P1 and P2 and loop 2 is between P2 and P3. The secondary structure has 18 conserved nucleotides as template and has 52 secondary bonds and 3 (C-C-U) repositioned nucleotides. The structure has one (G) inserted nucleotide in loop 1. Another ncRNA identified in the OA G20 genome, with two copies, was *Pseudomonas* P10. Both the structures OA G20_P10a and OAg20_P10b have 30 conserved nucleotides as template structure. Both structures OAg20_P10a and OAg20_P10b have 16 secondary bonds. No inserted or repositioned nucleotides were observed in either of the structures.

The OA G20 genome contains a gene for STnc490 (Hfq_binding RNA) sRNA. The secondary structure has four stems (P1-P2-P3-P4) and two loops—L3 and L4. The structure has 78 conserved nucleotides as reference template and 38 secondary bonds. Loop L3 has four inserted (U-U-A-A) and two repositioned nucleotides (A, A). The secondary structure of rnpB (RNase P type A) has 15 stems (P1–P2–P3–P4–P5–P6–P7–P8–P9–P10–P11–P12–P13–P14–P15), and 9 loops (L1–L3–L5–L7–L8–L10–L11–L12–L15). The secondary structure has 206 conserved nucleotides. It has 20 inserted and 26 repositioned nucleotides and 89 secondary bonds. Loop L1 has 14 inserted and 15 repositioned nucleotides. The secondary structure of hammerhead type II sRNA of OA G20 has threestems (P1-P2-P3) and one large loop (L2). The structure has 30 conserved nucleotides and 17 secondary bonds. Loop L3 has nine inserted nucleotides. The OA G20 genome contains one gene for tmRNA. The secondary structure of OAg20_tmRNA has eight stems (P1–P2–P3–P4–P5–P6–P7–P8) and six loops (L3–L4–L5–L6–L7–L8). It has a large central loop that connects five loops (L3–L4–L5–L6–L7). The secondary structure has 49 conserved nucleotides and 80 secondary bonds. It also has 25 inserted and 5 repositioned nucleotides.

The secondary structures of sX4 sRNA vary from one to another. The structure OAg20_sX4a has 14 secondary bonds, OAg20_sX4b has 10 secondary bonds, OAg20_sX4c has 13 secondary bonds, OAg20_sX4d has 15 secondary bonds, OAg20_sX4e has 15 secondary bonds, OAg20_sX4f has 14 secondary bonds, and OAg20_sX4g has 13 secondary bonds. There were multiple variations in the secondary structures compared to the template structure, in which 19, 22, 19, 18, 18, 18, and 19 nucleotides were inserted in structures a to f, respectively. Of the seven structures, OAg20_sX4a (1 nt) and OAg20_sX4f (2 nt) had repositioned nucleotides. In the case of conserved nucleotides’ position in the template, structure a has 20 (total 88) nucleotides, b has 20 (total 82) nucleotides, c has 18 (total 81) nucleotides, d has 23 (total 88) nucleotides, e has 23 (total 88) nucleotides, f has 10 (total 83) nucleotides, and g has 16 (total 83) nucleotides. The large variation in structures is reflected in the shape and size of loops and therefore in their functions.

## 4. Discussion

Genome-wide analysis for the prediction and identification of ncRNAs in the OA G20 genome revealed the presence of 37 ncRNAs (excluding tRNAs). The regulation of gene expression by riboswitches is well established and is a complicated RNA-based regulatory control in bacteria [45]. Riboswitches can switch in different conformations and regulate transcription. Riboswitches bind to a specific ligand, which leads to allosteric reorganization of mRNA structure, resulting in modulation of a specific gene or translation to a protein. Riboswitches are cis-regulatory elements and thus target genes involved in the same metabolic pathways, which leads to regulation through a negative-feedback loop [46]. Riboswitches from different classes have affinities for a large range of metabolites and coenzymes [47]. Riboswitches have been identified as potential targets for antibiofilm strategies [48].

The TPP riboswitch is one of the first identified regulatory elements at the 5′-UTR (untranslated region) of genes and is present in all domains of life, including bacteria [47]. TPP riboswitches regulate the genes responsible for the biosynthesis or transport of thiamine and thiamine pyrophosphate [49]. TPP, known as thiamine diphosphate (ThDP), is the biologically active state of thiamine (vitamin B_1_). ThDP and thiamine monophosphate (TMP) are the natural ligands for the TPP riboswitch [50]. Conformational change in TPP riboswitches also affects regulation and gene expression [51,52]. In the OA G20 genome, both the TPP riboswitches were found upstream of the gene *thiS*, involved in sulfate metabolism and transport. This gene is part of an operon, *thiSHFE*, responsible for thiamine biosynthesis and encodes sulfur carrier protein (ThiS), 2-iminoacetate synthase (ThiH), adenosyltransferase (ThiF), and thiamine phosphate synthase (ThiE) [53]. TPP binding with a TPP riboswitch will reduce the expression of ThiS protein, which may lead to low transport of sulfur in the cell and ultimately affect the biosynthesis of thiamine monophosphate and thiamine in OA G20. The OA G20 genome has another operon, *thiME*, which encodes for hydroxyethylthiazole kinase and thiamine monophosphate synthase, respectively, but no TPP riboswitch was found near this operon. This also suggests that TPP riboswitch regulation is critical in sulfur metabolism, cysteine metabolism, and thiamine biosynthesis in OA G20. The second TPP riboswitch was found upstream of the *thiM* gene (encodes for hydroxyethylthiazole kinase) in *D. vulgaris* Miyazaki F, *Erwinia cartovora*, *Rhodobacter spheroides* [54], and *E. coli* [51]. The TPP riboswitch has been identified as a potential target for antimicrobial compounds and reported in many pathogens as a drug target [55].

The cobalamin riboswitch is widely distributed in bacteria and already identified for more than 5000 bacterial species [56]. Cobalamin is the ligand for the cobalamin riboswitch. Cobalamin riboswitches are mostly specific to particular derivatives of cobalamin. The cobalamin could be active in two forms, such as 5′-deoxyadenosylcobalamin and methylcobalamin (MeCbl) [57]. Cobalamin riboswitches can be classified as class Cbl-I, Cbl-IIa, or Cbl-IIb. The genome of OA G20 has two cobalamin riboswitches, OA G20_cbla and OA G20_cblb, and both belong to class Cbl-I. The secondary structures of both riboswitches have a conserved core structure, but differ in the main helix and L4 (Appendix A). The first cobalamin riboswitch was found in the 5′-UTR of the gene *metE*, which encodes for 5-methyltetrahydropteroyltriglutamate–homocysteine methyltransferase and is involved in methionine biosynthesis. The second cobalamin riboswitch was found close to the 5′-UTR of the gene *cbikP*, which encodes for sirohydrochlorin cobaltochelatase CbiKP and anerobic cobalt chelatase and is a part of the cobalamin biosynthesis pathway. The second cobalamin riboswitch may be involved in the regulation of cobalamin biosynthesis, as reported in *D. vulgaris* (*cbikP* gene) [58]. Cobalamin (vitamin B_12_) plays a very important role in several enzyme functions, such as reductases, methylases, and deaminases, and is thus essential to all living cells. It has also been reported that bacteria have multiple cobalamin riboswitches to regulate gene expression. For example, *Desulfitobacterium hafniense* has 18 cobalamin riboswitches [59]. In *Listeria monocytogenes*, cobalamin riboswitches regulate the transcription factor PocR, which is related to biofilm formation [60]. Cobalamin riboswitches also regulate oxidative stress [61], salinity, and nitrogen stress [62]. Therefore, cobalamin riboswitches in OA G20 are involved in the regulation of methionine and cobalamin biosynthesis.

Bacteria have unique sensors for amino acid levels such as T-box elements and attenuators in mRNA, where T-box RNA binds tRNAs and modulates transcription or translation of the gene [63]. The glycine riboswitch is able to bind amino acids directly. Glycine riboswitches mostly involved in regulation of expression of genes for the glycine cleavage system, but were also reported upstream of several other genes responsible for the synthesis, conversion, or transport of glycine [64]. Only one glycine riboswitch was identified in the OA G20 genome, 5′ upstream of gene *alsT*, which encodes the amino acid carrier protein AlsT. The glycine riboswitch has been reported to control the cation efflux system in *S. pyogenes* [65] and glycine detoxification in *B. subtilis* [66]. The glycine riboswitch is involved in regulation of sodium:alanine symporter family protein and ultimately controls biofilm formation in *S. pyogenes* [65]. Although riboswitches regulate sulfur metabolism [67], no study has been undertaken on the role of glycine riboswitches in sulfur transport or metabolism in SRB.

The SAM-I riboswitch identified in the 5′-UTR of the mRNA of several bacteria. It regulates the expression of genes responsible for the biosynthesis of methionine (Met), cysteine, and SAM against the concentration of SAM [67]. The SAM-I riboswitch uses a negative-feedback mechanism to turn off Met biosynthesis and is very specific to its ligand molecule, SAM. SAM was reported to be involved in signaling and quorum sensing (QS), which ultimately leads to the synthesis of EPS and biofilm formation [68] and pathogenic interactions [69]. Only one SAM riboswitch was identified in the genome of OA G20, at the 5′-UTR of a gene (RS11395) that encodes for a hypothetical protein. This hypothetical protein may be involved in methionine synthesis, genome methylation, or sulfur metabolism in OA G20. We will explore its function in our future studies.

FMN riboswitches are widely present and reported in bacteria. They are involved in the regulation of the biosynthesis and transportation of riboflavin. Riboflavin is the precursor of flavin mononucleotide (FMN), and its cellular concentration regulates the conformational structure of the riboswitch [70]. The FMN riboswitch was reported at the 5′-UTR of the *ribD* operon, responsible for riboflavin biosynthesis in bacteria [71]. In the OA G20 genome, the FMN riboswitch was found at the 5′-UTR of the *ribB* gene, which encodes for 3,4-dihydroxy-2-butanone 4-phosphate synthase (DHBPS). DHBPS catalyzes the conversion of d-ribulose 5-phosphate (Ru5P) to l-3,4-dihydroxy-2-butanone 4-phosphate and formate in the riboflavin pathway. FMN regulates the operon with its presence and absence, (no FMN: transcription; FMN: premature termination). It has been reported that malfunction of the FMN riboswitch could inhibit bacterial growth, as riboflavin is involved in important cellular pathways. The selective inhibition of the FMN riboswitch inhibits growth in *E. coli* [72]. The role of the FMN riboswitch has been studied in *B. subtilis* using an analogue (roseoflavin) of FMN and riboflavin, and roseoflavin downregulated the expression of the gene [73]. Riboflavin and FMN are both categorized as soluble electron transfer mediators and important for growth of SRB and its biofilm. A study shown that flavin-like molecules were involved in electron transfer in OA G20 biofilm and corrosion on carbon steel surfaces [74]. Recent studies identified that exogenous riboflavin can induce biofilm formation in *Shwenella oneidensis* [75] and increase corrosion on the nickel surface of a *D. vulgaris* biofilm [76]. Therefore, the FMN riboswitch could be involved in regulation of biofilm formation in OA G20. C-di-GMP has been identified as an important signaling molecule that controls and regulates various cellular functions, such as virulence, motility (motile to sessile), biofilm formation and dispersion, flagellar motility, adhesion to surfaces, cell–cell signaling and cell differentiation [77]. C-di-GMP is involved in bacterial exopolysaccharide synthesis and biofilm formation in many bacterial species, such as *B. subtilis* and *P. aeruginosa* [78,79]. C-di-GMP performs its regulatory function by binding to a diverse group of effectors, such as kinases or phosphorylases [80], transcription factors [81], PilZ domain proteins [82], and riboswitches [83]. C-di-GMP riboswitches are mostly located in the 5′-UTRs of genes responsible for motility and biofilm formation, and regulate genes in response to the change in concentration of c-di-GMP [48]. C-di-GMP-I binds to c-di-GMP and regulate the expression of downstream genes [81]. The role of c-di-GMP-I in adaptation and virulence through the collagen adhesion protein is important for biofilm formation in *B. thuringiensis* [84]. An analysis identified only one c-di-GMP-I riboswitch in the OA G20 genome, whereas more than one c-di-GMP riboswitch has been reported in other bacteria [85]. The c-di-GMP-I riboswitch was found at the 5′-UTR of gene *norR* in the OA G20 genome, which encodes the anaerobic nitric oxide reductase transcription regulator NorR. The OA G20 genome has 13 copies of the *norR* gene and therefore has a very significant role. *NorR* is a NtrC/Nif family regulator and involved in nitrosative stress protection against reactive nitrogen species (RNS) in *D. gigas* [86] and *E. coli* [87]. A mutation study of c-di-GMP-I in *D. vulgaris* Hildenborough established its role in the modulation of response regulators, as the mutant lost its biofilm formation ability [88]. NorR is a multifunctional regulator and regulates expression of genes for extracellular toxins, enzymes, and cell surface-associated protein in *Staphylococcus*. A higher number of norR genes in the OA G20 genome indicates its involvement in many physiological activities. Unfortunately, no study is available on noR regulation in OA G20.

Selenocysteine was first reported in protein A of glycine reductase in *C. sticklandii* [89]. tRNA-Sec (selenocysteine transfer RNA) is a unique tRNA that plays a very important role in the synthesis of L-selenocysteine [90]. Sec is considered the 21st amino acid (“stop codon-UGA”) and is structurally like cysteine, but selenium replaces sulfur. This change enhances the catalytic and redox properties of selenocysteine [91,92]. tRNA-Sec regulates the synthesis of selenocysteine, which plays crucial role in the oxygen resilience (reducing reactive oxygen species) of OA G20. In our other study, using a text-mining and protein–protein network approach, we found a gene *SelA* (synthesizes Sec from Ser-tRNA) in the OA G20 genome to be essential [42]. Researchers have identified a selenocysteine-containing enzyme (formate dehydrogenase H) in *E. coli* [93]. OA G20 is a strict anaerobe, and previous studies suggest that anaerobic microorganisms are rich in selenoproteins [94]. The role of selenoproteins has been reported in bacterial pathogenesis and biofilm formation [95]. Xanthine dehydrogenase is another selenium-dependent enzyme that has been reported to support biofilm formation in *Enterococcus faecalis* [96]. This finding suggested that tRNA-Sec plays a very significant role in cellular physiology and biofilm formation mechanisms of OA G20.

Bacterial small SRP is a protein–RNA complex that binds ribosomes translating secretory and membrane proteins and the SRP receptor (SR). Regulatory RNAs involved in translation include ffs (RNA component of SRP for co-translational translocation of new proteins across membranes) and SsrA (tmRNA) for regulation of stalled ribosomes [97,98]. SRP RNA (Ffs), also known as 4.5S, is one of the sRNAs functioning as a component of SRP. SRP RNAs are involved in mRNA localization on membrane for membrane proteins in *E. coli* and *Lactobacillus lactis* [99,100]. Ffs RNA has been reported as extracellular RNA for intercellular communication in biofilms or mixed communities of *E. coli* [101], and *Salmonella enterica* [102]. The OA G20 genome has one ffs RNA located between tRNAs in the genome and its regulatory function is unknown. TmRNA (SsrA RNA; 10Sa), or transfer-messenger RNA, is a type of sRNA conserved among bacteria encoded by the *ssrA* gene. TmRNA has both a tRNA and mRNA-like domain [103]. Only the third analysis approach resulted in one tmRNA in the OA G20 genome. TmRNA facilitates *trans*-translation, which rescues stalled ribosomes. TmRNA acts as both tRNA and mRNA [104] and also influences physiological bioprocesses in some bacteria [105]. A recent study identified two genes, *acoA* and *yhjR*, as part of the tmRNA regulation system responsible for biofilm formation in *B. subtilis* [105]. Currently, no study on the role of tmRNA regulation is available in SRB.

The 6S (SsrS) RNA was first identified in *E. coli* encoded by the *ssrS* gene [106], and is one of the best-known small prokaryotic ncRNAs, widely reported in all branches of bacteria and a key regulator of transcription. It has the unique ability of acting as a transcription template for the synthesis of short product RNAs (pRNAs). It has high affinity for σ70-RNA polymerase (RNAP) holoenzyme (Eσ70). It hinders binding of many DNA promoters to RNAP and inhibits transcription from σ70-responsive promoters of the majority of genes [107]. It has been reported to play a role in regulation of a signaling molecule (guanosine tetraphosphate) responsible for stress responses, oxidative stress response, and growth adaptation in *E. coli* [108]. Regulation dependent on 6S RNA in transcription has been reported to be involved in pH tolerance [109]. We have observed the pH homeostasis ability of OA G20 in our lab (unpublished) and 6S RNA could play a role here in helping bacterial cells to optimize pH for growth. The OA G20 genome has a single 6S RNA, and we will decode its role and mechanism of regulation in our future studies.

RnpB *(RNase P type A):* RNase P is an essential catalytic RNA (ribonucleoprotein enzyme), encoded by the *rnpB* gene that processes pre-tRNA (ptRNA) gene transcripts to yield mature tRNAs (mtRNA). *E. coli* and *B. subtilis* have different types of RNase P: type A and type B [110]. Like *E. coli*, OA G20 has the type A RNase P. The expression of *rnpB* was regulated by presence and absence of arabinose [111], but no such study is available on SRB. The hammerhead ribozyme (HHR) was first reported in 1986 [112] and is a small endonucleolytic ribozyme. HHR is a self-cleaving RNA that catalyzes the scission of its own phosphodiester backbone. HHRs are mostly identified as type I and type III, and a motif-based sequence search discovered type II HHR in bacteria recently in 2011. HHRs has now been reported to have diverse origin of metagenomic data, suggesting its widespread presence in viruses, prokaryotes, archaea, and bacteriophages [112,113]. This study is the first report on the HHR type II ribozyme in any SRB (OA G20), and its function needs to be validated and explored in OA G20 in further research.

*STnc490 Hfq-binding RNA:* The RNA chaperone protein Hfq is a key post-transcriptional regulator in bacteria, an sRNA that binds the bacterial RNA-binding protein Hfq known as Hfq-binding sRNA. Previous studies identified a total of 64 Hfq-binding sRNAs in bacteria [114,115]. This current study led to the identification of only one Hfq-binding RNA (STnc490) in the OA G20 genome. A similar genome-wide study identified 40 candidate Hfq-dependent sRNAs in *Erwinia amylovora* [115,116]. Hfqs were reported for negative regulation in bacteria [114,115,117]. Hfq-dependent regulation of *rpoS* translation has been reported in *E. coli* and *Xanthomonas campestris* [116,118], confirming its role in translational control. sX4 is a type of sRNA first identified in *Xanthomonas* [119,120]. This study identified seven sX4 sRNAs in the OA G20 genome. These seven sX4 sRNA have six different secondary structures, suggesting their involvement in distinct regulatory mechanisms. This is the first report where sX4 sRNAs have been identified in an SRB genome such as OA G20. *Pseudomonas* sRNA P10 (RF01668) is ncRNA was identified in *Pseudomonas* [120]. This sRNA was found conserved across most of the *Pseudomonas* species, but its functions and role have yet to be determined. This study identified two sequences of P10 sRNAs in the OA G20 genome. This is also the first report on the prediction and identification of P10 sRNAs in any SRB. The functions of P10 sRNAs in SRB (OA G20) cellular mechanisms and pathways need to be elucidated.

In summary, the present study identified 37 ncRNAs in the OA G20 genome, excluding tRNAs (numbers varied: 58–66). This study identified five ncRNA (sRNAs)—*Pseudomonas* P10, Hammerhead type II, sX4, tmRNA, and tRNA-Sec—that have not been detected or reported in any SRB. Of these five, three are rare—P10, hammerhead type II, and sX4—and were only identified in few species. The predicted riboswitches (TPP, SAM, and cobalamin) have unique sequences and could be explored for their functional efficiencies in structural simulation studies further. The identified ncRNAs play critical roles in the regulation of mechanisms and pathways involved in biosynthesis, membrane transport, signaling, quorum sensing, motility, response, sulfur metabolism, and anaerobic stress regulation. All these physiological activities are key for growth, survival, and biofilm formation in OA G20. This is the first study to identify the genome-wide regulome of OA G20, and these findings will lead to research elucidating the regulatory mechanisms involved in SRB and biofilm formation, ultimately fulfilling the goal of understanding the “rules of life” of SRB.

## 5. Conclusions

This is the first genome-wide computational analysis of ncRNAs identified in the SRB OA G20. Different approaches resulted in new ncRNAs/sRNAs, which suggests that the prediction and identification of ncRNAs depends on the methods and algorithms best for genome organization, with scope remaining to develop new methods. These results suggest that there is a huge probability of identifying novel ncRNAs from the OA G20 genome. We will extend this study further by applying other computational methods and small RNA-sequencing analysis, followed by expression validation, functional role assignment, and network analysis.

## Figures and Tables

**Figure 1 microorganisms-12-00960-f001:**
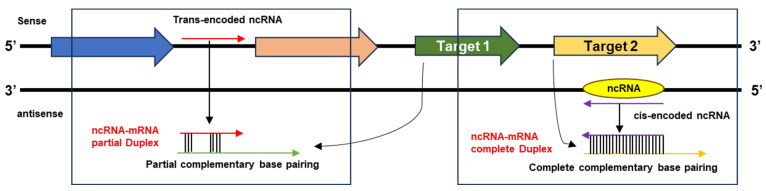
Origin of trans-encoded and cis-encoded ncRNAs from sense and antisense strands. Trans-encoded ncRNAs (Red arrow) have partial base pairing with target mRNAs (Green arrow) and form partial duplexes. Cis-encoded ncRNAs (Purple arrow) form complete duplexes with target mRNAs (Yellow arrow).

**Figure 2 microorganisms-12-00960-f002:**
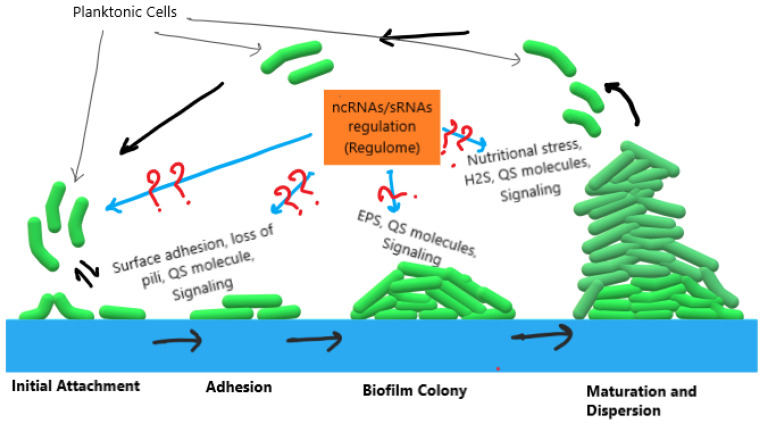
Proposed knowledge gaps regarding the regulome of OA G20 biofilm formation.

**Figure 3 microorganisms-12-00960-f003:**
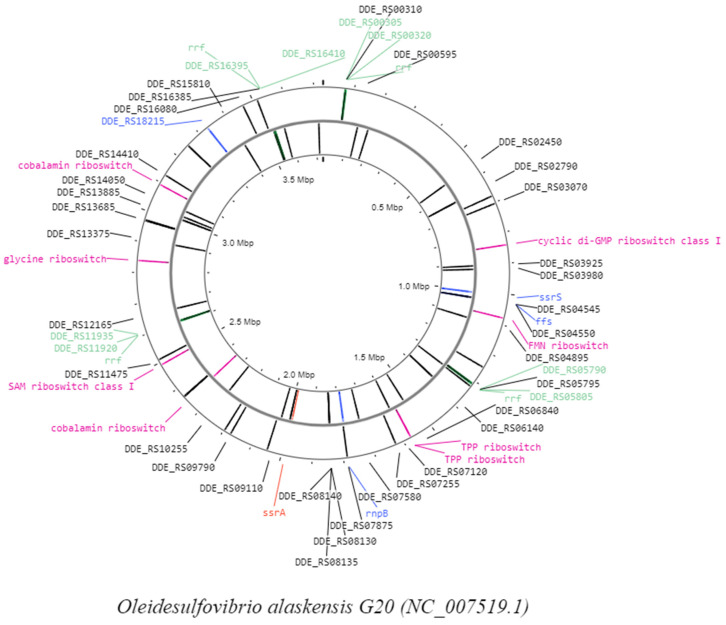
ncRNA map of OA G20 genome generated by Proksee server. The gene IDs with different colors represent categories of predicted ncRNAs where black—tRNA; green—rRNAs; purple—regulatory RNAs; blue—ncRNAs; and red—tmRNA.

**Table 1 microorganisms-12-00960-t001:** Details of software and tools used in genome-wide identification of ncRNAs in OA G20.

Approaches	Genome Sequence	Genome Reso. Acc.	Software and Tools
Approach 1	3,730,232 bp	NC_007519.1 NCBI	Rfam, cmsscan, infernal, R2DT, RNAcentral
Approach 2	3,730,232 bp	CP000112.1 ENA/EMBL	Rfam, R2DT, RNAcentral
Approach 3	3,730,232 bp	NC_007519.1 NCBI	Proksee server, R2DT, RNAcentral

NCBI: National Center for Biotechnology Information; ENA: European Nucleotide Archive, EMBL: European Molecular Biology Laboratory.

**Table 2 microorganisms-12-00960-t002:** Annotated genomic features of OA G20.

Gene Types	
Protein coding	3257
tRNAs	66
rRNAs	12
Pseudogenes	37
Miscellaneous RNAs	5

**Table 3 microorganisms-12-00960-t003:** ncRNAs identified in OA G20 genome using approach 1.

ncRNA Family	Rfam ID	Gene Type	Length	Genome Location	Strand	Description
5S_rRNA	RF00001	rRNA	119	3,533,731:3,533,617	−	5S ribosomal RNA
Glycine_riboswitch	RF00504	Cis-reg; riboswitch	94	2,838,359:2,838,468	+	Glycine riboswitch
TPP_riboswitch	RF00059	Cis-reg; riboswitch	105	1,573,919:1,574,022	+	TPP riboswitch (THI element)
TPP_riboswitch	RF00059	Cis-reg; riboswitch	105	1,574,080:1,574,186	+	TPP riboswitch (THI element)
Cobalamin_riboswitch	RF00174	Cis-reg; riboswitch	189	3,093,545:3,093,727	+	Cobalamin riboswitch
SAM riboswitch	RF00162	Cis-reg; riboswitch	108	2,490,976:2,491,083	+	SAM riboswitch (S box leader)
SSU_rRNA_bacteria	RF0177	rRNA	1533	2,609,189:2,607,648	−	Bacterial small subunit ribosomal RNA
SSU_rRNA_archaea	RF01959	rRNA	1477	1,309,443:1,310,984	+	Bacterial small subunit ribosomal RNA
SSU_rRNA_microsporidia	RF02542	rRNA	1311	3,538,761:3,537,220	−	Bacterial small subunit ribosomal RNA
SSU_rRNA_eukarya	RF01960	rRNA	1831	69,852:71,393	+	Bacterial small subunit ribosomal RNA
LSU_rRNA_bacteria	RF02541	rRNA	2925	2,607,209:2,604,279	−	Bacterial large subunit ribosomal RNA
LSU_rRNA_archaea	RF02540	rRNA	2987	3,536,781:3,533,851	−	Bacterial large subunit ribosomal RNA
LSU_rRNA_eukarya	RF02543	rRNA	3401	71,832:74,762	+	Bacterial large subunit ribosomal RNA
tRNA_Sec	RF01852	tRNA	91	1,531,711:1,531,804	−	tRNA-Sec
tRNA	RF00005	tRNA	73	2,047,102:2,047,175	+	tRNA

**Table 4 microorganisms-12-00960-t004:** Comparison of ncRNAs identified in the OA G20 genome using approaches 2 and 3.

ncRNA Family	Rfam IDs	Gene Type
Identified in approaches 2 and 3		
Glycine_riboswitch	RF00504	cis-reg; riboswitch
TPP_riboswitch	RF00059	cis-reg; riboswitch
TPP_riboswitch	RF00059	cis-reg; riboswitch
FMN_riboswitch	RF00050	cis-reg; riboswitch
c-di-GMP-I	RF01051	cis-reg; riboswitch
Cobalamin_riboswitch	RF00174	cis-reg; riboswitch
Cobalamin_riboswitch	RF00174	cis-reg; riboswitch
SAM-Box	RF00162	cis-reg; riboswitch
Bacterial small_SRP	RF00169	gene
6S	RF000013	SsrS_RNA
5S_rRNA	RF00001	rRNA
5S_rRNA	RF00001	rRNA
5S_rRNA	RF00001	rRNA
5S_rRNA	RF00001	rRNA
SSU_rRNA_bacteria	RF0177	rRNA
SSU_rRNA_bacteria	RF0177	rRNA
SSU_rRNA_bacteria	RF0177	rRNA
SSU_rRNA_bacteria	RF0177	rRNA
LSU_rRNA_bacteria	RF02541	rRNA
LSU_rRNA_bacteria	RF02541	rRNA
LSU_rRNA_bacteria	RF02541	rRNA
LSU_rRNA_bacteria	RF02541	rRNA
tRNA_Sec	RF01852	tRNA
Only (unique) ncRNAs identified from approach 2		
sX4	RF02223	sRNA
sX4	RF02223	sRNA
sX4	RF02223	sRNA
sX4	RF02223	sRNA
sX4	RF02223	sRNA
sX4	RF02223	sRNA
sX4	RF02223	sRNA
P10	RF01668	sRNA
P10	RF01668	sRNA
STnc490	RF01405	sRNA
tRNA_Sec	RF01852	tRNA
Only (unique) ncRNAs identified from approach 3		
tmRNA	RF00023	SsrA
rnpB	RF00010	rnpB
DDE_RS18215	RF02276	hammerhead

## Data Availability

The whole-genome sequence data of OA G20 used in the study can be found on NCBI (https://www.ncbi.nlm.nih.gov/search/all/?term=NC_007519.1 (8 January 2023)). Results files are given in Appendix A.

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
