# Peer review of "Genome-Wide Computational Prediction and Analysis of Noncoding RNAs in *Oleidesulfovibrio alaskensis* G20"

_microorganisms, 2024, doi:10.3390/microorganisms12050960_

Round 1

Reviewer 1 Report

Comments and Suggestions for Authors

Genome Wide Computational Prediction and Analysis of Noncoding RNAs in 2 Oleidesulfovibrio alaskensis G20

Ram Nageena Singh and Rajesh K. Sani 

General comments.

The authors present a genomic analysis of  Oleidesulfovibrio alaskensis G20 (OA G20) using three putatively different approaches to identify noncoding RNAs (ncRNAs) that have regulatory roles in cellular metabolism. They are particularly interested in the regulation of biofilm formation in OA G20. The identified ncRNAs are presented in the results section where the authors enumerate all identified ncRNAs from each method. In the discussion the authors again list the identified ncRNAs that are probably, based on sequence analysis, regulatory elements. 

Regulatory small RNAs are certainly an interesting topic and focusing on an economically important sulfate reducing bug like OA G20 is logical and might be of interest to the field. There are, however, some confusing aspects of the study. The authors make the claim that they are using three different approaches to screening the OA G20 genome. In the first approach the authors download and analyze the genome of NC_007519.1 using the online search available through the Rfam server. In the second method the authors select the genome of OA G20 (CP000112.1) from the Rfam server and the ncRNAs previously identified by Rfam were used for "further" analysis. This could be confusing to the reader inasmuch as NC_007519.1 and CP000112.1 appear to be identical. Moreover why these approaches would yield  different results is not clear and why the authors selected these two apparently identical methods is not made clear. Table 1, in which the tools used for screening are identified, indicate that the first method includes cmsscan and infernal while the second method does not. Moreover there was a substantial difference in the number of ncRNAs detected, 34 versus 93 for the first and second methods respectively. Why is there such a substantial difference in these two methods given that the genomes are the same and the Rfam server is used in both cases? It would appear that all the hard work has already been done by Rfam and that the first method provided little in the way of additional data. There is not a clear statement in the text regarding how many ncRNAs were detected using their third approach using Proksee but reviewing supplemental Table 2 suggests that the third approach was not as sensitive as approaches 1 and 2. In their analysis the authors go through all the ncRNA riboswitches that were detected in each approach, hence there is a considerable amount of repetition in the results section.  A more logical approach would be to quantify the number of riboswitches  using each method of detection and then systematically describe those ncRNA riboswitches common to all 3 or to two approaches and those that are unique to a single detection method. There was no critical discussion on the possible reasons for the differences seen in the three difference approaches and which the authors feel is preferred. Why eukaryotic and archaeal rRNAs were found in the first method but not the second or third methods is mysterious. In addition to the repetition found in the results section, the discussion is too long. Their interest is in the detection of riboswitches in OA G20 and how these regulatory elements might impact biofilms. It would be helpful to the reader if the authors clearly identified riboswitches that they could logically posit had a role in biofilms. 

There are also a considerable number of awkward sentences, incomplete sentences, missing articles, incorrect tense and confounding syntax. I am sure that I have not detected all of the grammatical problems. The paper should be carefully re-proofed before resubmission. Reorganizing and reducing repetition would considerably enhance the manuscript.

Specific comments.

L17  Incomplete sentence

L33-34  This sentence is a bit confusing. Needs to be recast

L41  "stochastic variations" Needs a better description. Do you mean variations in abundance?

L76  Sentence not clear. Needs revising.

L103-104  Poorly phrased.

L120  "splitfasta"  Reference needed

L175. Why do you find eukaryotic and archaeal rRNAs? Please explain presence of these rRNAs.

L192  Re. SAM riboswitches, reference?

L193-194  Sentence should be revised.

L196-197  Remove italics.

L198  Change to "applying the second approach"

L219  enrichment? Be more concise. Eg. This approach resulted in the identification of more ncRNAs such as FMN riboswitch, c-di-GMP-I riboswitch, P10, STnc490, sX4, 6S and Bacterial small SRP than the first approach.

L221. Again, I would not consider this an enrichment, it would appear to be a refined analysis leading to more sensitivity in identification.

Table 3. I note the complimentary tables in the supplemental but some type of summary table in whch the reader can easily compare the identifications common to all three techniques as well as those that are unique to a method.

L213  The number of riboswitches identified was the same as in the second approach.

L236-245  Should this detail be here or in the discussion? I note that details for other RNAs appear in the discussion.

L332-343  The section that discusses the proposed secondary structures of the RNAs is quite tedious. If the authors feel that this section is absolutely necessary then OK. However I urge the authors to somehow condense or provide a better synthesis to avoid the tedium that could be conveyed admirably by figures.

L350-360  Much of this is a repeat of the intro.

L361  This is not clear. Should be recast. UTR has not been defined yet.

L388  of the gene metE which encodes for...

L389  and is involved in...

L362-363  Poorly structured sentence

L370-374  Importantly, how do you posit these TPP riboswitches integrate into the metabolism of OA G20. Be specific here, if thiamine phosphate binds to the TPP riboswitch, what are the consequences regarding expression.

L397  D. haf  Font size

L388-392  Similar to my comment on the TPP switch, how do you perceive these cobalamine switches regulating metabolism in OA G20?

L427. Be concise. "Only one SAM-riboswitch was identified in the genome of  OA G20...""

L428-430  Poorly structured sentence. Needs to be recast.

L471-475  This is of interest and requires somewhat more explanation. You note that there is only one c-di-GMP-I riboswitch in OA G20 yet there are 13 copies of norR genes. How do you perceive this regulatory scheme? What is the genome positions of the 13 copies?

L529-531  This sentence is particularly confusing. Requires recasting.

L519-520 The lead up to this sentence could be shortened considerably given the uninformative conclusion.

L507-508  rather vague

L586-589  Awkward sentence. Needs to be recast.

Comments on the Quality of English Language

See above

Author Response

Thanks for the review and critical suggestion of our manuscript. We have revised the manuscript as per the suggestions to improve it.

Pls see the attachment. 

Reviewer 2 Report

Comments and Suggestions for Authors

In this study, the authors identify ncRNAs in the genome of a model SRB, Oleidesulfovibrio alaskensis G20 (OA G20), which could play key roles in the regulation of metabolic pathways and transport etc. The manuscript was acceptable after revision. The major weak points are that conventional prediction and analysis methods were used and the novelty was not obvious. The specific comments are as follows:

(1)   The genome location of all the ncRNAs discussed in section 4 are better given.

(2)   In the introduction section (lines 106-107), the aim of this study was to identify the ncRNAs related with biofilm formation. While the ncRNAs investigated were not limited to biofilm formation. Actually, the mechanism of biofilm formation as well as its regulation were not clear enough.

(3)   There are many type errors, uneven font styles and sizes etc.

Comments on the Quality of English Language

The English was acceptable.

Round 2

Reviewer 1 Report

Comments and Suggestions for Authors

The authors have addressed all of my comments from the review. From my perspective, the paper is ready for publication.